# Vitamin D Receptor Influences Intestinal Barriers in Health and Disease

**DOI:** 10.3390/cells11071129

**Published:** 2022-03-27

**Authors:** Jun Sun, Yong-Guo Zhang

**Affiliations:** 1Department of Medicine, University of Illinois at Chicago, Chicago, IL 60612, USA; yongguo@uic.edu; 2Department of Microbiology/Immunology, University of Illinois at Chicago, Chicago, IL 60612, USA; 3Jesse Brown VA Medical Center Chicago (537), 820 S Damen Ave, Chicago, IL 60612, USA

**Keywords:** vitamin D, VDR, tight junctions, cancer, COPD, inflammation, infection, microbiome

## Abstract

Vitamin D receptor (VDR) executes most of the biological functions of vitamin D. Beyond this, VDR is a transcriptional factor regulating the expression levels of many target genes, such as genes for tight junction proteins claudin-2, -5, -12, and -15. In this review, we discuss the progress of research on VDR that influences intestinal barriers in health and disease. We searched PubMed and Google Scholar using key words vitamin D, VDR, tight junctions, cancer, inflammation, and infection. We summarize the literature and progress reports on VDR regulation of tight junction distribution, cellular functions, and mechanisms (directly or indirectly). We review the impacts of VDR on barriers in various diseases, e.g., colon cancer, infection, inflammatory bowel disease, and chronic inflammatory lung diseases. We also discuss the limits of current studies and future directions. Deeper understanding of the mechanisms by which the VDR signaling regulates intestinal barrier functions allow us to develop efficient and effective therapeutic strategies based on levels of tight junction proteins and vitamin D/VDR statuses for human diseases.

## 1. Introduction

The intestinal epithelium cells (IECs) are essential in barrier function, structural function, and host defense. The tight junctions (TJs) seal the space between adjacent epithelial cells. TJs are made by a number of proteins, including claudins, occludin, junctional adhesion molecules (JAM), and tricellulin and cytoplasmic plaque proteins, e.g., zonula occludens (ZO) proteins. Interestingly, VDR is known to transcriptionally regulate several genes for TJs, including claudin-2, -5, -12, and -15 in the intestinal epithelial cells [1,2,3,4]. VDR deletion in IEC cells leads to decreased protein levels of claudins, such as claudin-2 and -12, which contribute to vitamin D-dependent homeostasis. However, VDR regulation of some TJ proteins may be vitamin D-independent.

In 2013, we reviewed the roles of vitamin D and VDR in tissue barriers [5]. Back then, our understanding of VDR in influencing intestinal barriers was limited to its role as a transcriptional factor and a receptor for vitamin D. We did not have sufficient evidence on its tissue-, gender-, and time-specificity in regulating barrier function. We did not sufficiently consider the physiological relevance in vivo and in disease models. We did not know that the human gut microbiome is determined by the variation of *vdr* gene [6]. Indeed, ~3% of the mouse and human genomes are regulated directly or indirectly by vitamin D, suggesting widespread effects of vitamin D/VDR in various disease mechanisms [7,8,9]. Thus, the traditional model of treating cultured cells with vitamin D_3_ is now giving way to models with a more complex mechanism of action.

In the current review, we discuss the progress of research on VDR that influences intestinal barriers in health and disease. We searched PubMed and Google Scholar using key words “vitamin D, VDR, tight junctions, cancer, inflammation, and infection”. We summarize studies related to VDR/TJ distribution, cellular functions, and mechanisms; we discuss the role of VDR and barriers in various diseases, e.g., IBD, colon cancer, chronic obstructive pulmonary disease (COPD), infection, and other diseases, mainly focusing on the intestinal diseases. We also discuss the limits of current studies and future directions. New insights into VDR and TJs will allow us to better understand the pathophysiology of diseases and develop novel strategies for prevention and treatment.

### Intestinal TJs for Barrier Functions

The intestinal barriers are provided by several layers: mucus, epithelial cells and TJs, and the local immune system. The intestinal mucus layer has a primary role in host protection against mechanical, chemical, and biological injuries. The intestinal epithelium is the second layer, which serve as a barrier, provide structure, and play a role in host defense [10,11,12,13,14,15,16]. TJs seal the space between adjacent epithelial cells and regulate epithelial polarity and vectorial movement of solutes and fluids in the intercellular space. The microbiome contributes to maintaining the integrity of the intestinal epithelium. Bacteria can induce proliferation of IECs, which is evidenced by lower proliferative IECs in germ-free (GF) than those in conventional mice [17]. A recent study [18] showed evidence on microbiota and their product butyrate in regulating expression of actin-binding protein synaptopodin for intestinal homeostasis. Synaptopodin is localized to the TJs and within F-actin stress fibers of intestinal epithelial cells. It is critical for barrier integrity and cell motility. Butyrate regulation of synaptopodin reveals a direct mechanistic link between microbiota-derived metabolite and intestinal barrier functions.

TJ structural components, which determine epithelial polarization and intestinal barrier function, can be regulated by homeodomain proteins that control the differentiation of the intestinal epithelium [19,20]. TJ proteins include claudins, occludin, JAM, and ZO. Many TJ proteins, such as ZO and occludin, tighten cell structure and maintain a barrier [21]. Some TJ proteins, such as claudin-2, are considered as “leaky” proteins because they form paracellular water channels [22] and increased their expression in inflammation and cancer [23,24,25,26,27,28]. Mikio Furuse was the first to purify and identify claudins in the laboratory of the late Shoichiro Tsukita [29]. The name is taken from the Latin word “claudere,” meaning to close, because it was anticipated that these proteins might constitute the tight junctional barrier [30]. *Occludin*, *tricellulin*, and *JAMs* genes exist as a limited number of different isoforms; they are unlikely to account for the diversity of paracellular permeability characteristics in different epithelia [30]. There are 27 claudin family members that contribute to tight junctions [31], and not all claudins are the same. Thus, many studies are on claudins to determine paracellular selectivity. Some claudins have other roles in cell signaling and differentiation. For example, a recent study reported that claudin-7 controls intestinal crypt stem cell survival, self-renewal, and epithelial differentiation through Wnt/β-catenin signaling in mice [32]. claudin-2 and claudin-12 form paracellular Ca^2+^ channels in human intestinal epithelial cell lines [1].

In a healthy adult intestine, the expression and apical tight junction localization of zonula occludens 1 (ZO-1) and occludin are unchanging. In contrast, most claudins are expressed in decreasing or increasing gradients or in more complex patterns along the longitudinal axis of the intestine and the crypt to villus/surface differentiation axis, based on the observations in mice [33]. During neonatal development, several claudins altered in transcript expression as well as changes in localization along the crypt–villus axis. claudin-19 was only detected at days 1 and 14 in mice [33]. In Figure 1, claudin-2 restricts to the crypt at the basal lateral side of the mouse intestine, whereas claudin-7 localizes in the epithelial cell membrane along the crypt–villus axis, based on our previous reports [2,27]. From the website Protein Atlas, we can find some claudins related to their distribution, expression levels in different human organs, disease-related information, and validated antibodies for research (https://www.proteinatlas.org/ENSG00000184113-CLDN5, accessed on 2 February 2022).

## 2. Vitamin D/VDR and TJs in Intestinal Homeostasis

Classically, 1,25-dihydroxy vitamin D (1,25(OH)_2_D_3_), the active form of vitamin D, plays a key role in calcium homeostasis and in electrolyte and blood pressure regulation [34]. VDR mediates most known physiological functions of 1,25(OH)_2_D_3_. Once VDR binds with 1,25(OH)_2_D_3_, VDR heterodimerizes with the retinoid X receptor (RXR) in nuclei, then binds to the vitamin D-response element (VDRE) in the promoter of the target gene, thus activating gene transcription [35]. VDR target genes include genes for the antimicrobial peptide (AMP) cathelicidin precursor (LL-37) [36], β defensin [37], autophagy regulator autophagy related 16 Like 1 (ATG16L1) [38], and proliferation regulator Axin1 [39], in addition to the genes for *claudin-2*, *-5*, *-12*, and -*15* [1,2,3,4].

VDR is critical in regulating intestinal homeostasis by preventing pathogenic bacterial invasion, inhibiting inflammation, and maintaining barrier functions [36,40,41,42,43,44]. In the Protein Atlas (https://www.proteinatlas.org/ENSG00000111424-VDR, uploaded on June 2018; accessed on 23 February 2022), we can find that the presence of VDR in various human and mouse tissues, along with its ability to exert differentiation [45,46], growth, and inhibitory [46] and anti-inflammatory actions [43,47,48,49], sets the stage for therapeutic exploitation of VDR ligands for the treatment of various inflammatory conditions. VDR deletion in human IEC cells lead to decreased claudin-2 and -12 [1]. However, little is known about how the intestinal VDR are involved in TJs and intestinal inflammation and infection, especially in in vivo systems.

In Table 1, we summarize the studies on vitamin D/VDR directly or indirectly influence the expression and function of cell junction proteins in different tissues from human and mouse samples. Although the primary focus of this review is on the intestinal epithelial TJ barrier, the studies in other cell types, including lung, skin, kidney, cornea, brain, oral, and urinary bladder, are also included in the current table.

## 3. Novel Roles of Vitamin D, VDR, and TJs in Diseases

Emerging evidence demonstrates that vitamin D/VDR deficiency is a critical factor in the pathology of many diseases, such as IBD and colon cancer, among others [45,74,75,76,77,78,79,80,81]. In the following section, we discuss the tissue-specific regulatory functions of VDR on TJ proteins in different diseases, based on clinical data and laboratory evidence.

### 3.1. Vitamin D, VDR, and TJs in Intestinal Inflammation

Multiple factors (e.g., risk genes, environmental triggers, immunity, and microbiome) contribute to the pathogenesis of IBD [16,82,83,84,85]. Susceptibility to IBD is associated with polymorphisms in the *vdr* gene [86,87,88,89,90]. Low vitamin D status and reduced VDR is observed in patients with IBD [91,92,93,94]. Vitamin D influences the course and severity of human IBD [92]. Vitamin D/VDR appears to be an important immunological regulator of IBD [94,95]. In experimental models, VDR knockout (KO) mice spontaneously developed colitis [96,97,98,99,100,101] and VDR/IL-10 double KO mice developed severe IBD [102]. Our studies further demonstrated that mice with conditional deletion of intestinal epithelial VDR are susceptible to colitis, due to dysbiosis, dysfunction of Paneth cells, dysregulated autophagy, and disrupted TJs [27,38].

Luminal stimuli and epithelial cell dysfunction are known to contribute to IBD pathogenesis and progression [103,104]. Defective epithelial barrier function has been implicated in IBD and can predict relapse during clinical remission. Increased permeability is also present in a subset of unaffected first-degree relatives of patients with Crohn’s disease [105], a type of IBD [104]. Claudin-2 forms a paracellular water channel and thus mediates paracellular water transport in leaky epithelia [22,33,106,107,108]. Changes in the expression and distribution of claudin-2, -5, and -8 lead to discontinuous TJs and barrier dysfunction in active CD [26,106]. Loss-of-function mutations in protein tyrosine phosphatase nonreceptor type 2 [109] increase the risk of IBD and celiac disease. Interestingly, T-cell protein tyrosine phosphatase (TCPTP) protects against intestinal barrier dysfunction induced by the inflammatory cytokine IFN-gamma. The mechanisms are that it maintained localization of ZO-1 and occludin at apical tight junctions and restricted both expression and insertion of the cation pore-forming transmembrane protein, claudin-2, at TJs through upregulation of matriptase, the inhibitory cysteine protease, in experimental mouse models [109].

Intestinal epithelial VDR deletion led to decreased claudin-2 at both mRNA and protein levels in mice [2]. Previous studies used the whole VDR^−/−^ mouse model, which is known to have problems in various organs. To determine the role of intestinal epithelial VDR and mechanisms by which VDR regulates claudin-2, intestinal epithelial-specific VDR knockout VDR^∆IEC^ mice were used. It was demonstrated that claudin-2 is transcriptionally regulated by VDR in the healthy intestine; however, claudin-2 is hyper regulated through inflammatory signaling in colitis with reduced intestinal epithelial VDR [2,27].

Claudin-15 is significantly lower in patients with IBD, compared to the health controls [3,110]. VDR conditional knockout VDR^∆IEC^ or overexpressing mouse models, cultured human cell lines, and organoids were used to investigate the mechanisms of altered claudin-15 in intestinal inflammation [3]. Overexpression of intestinal epithelial VDR resulted in significantly increased claudin-15 and decreased susceptibility to chemically and bacterially induced colitis. Colonic claudin-15 was reduced in VDR^∆IEC^ mice, which were susceptible to colitis. Direct binding of VDR to the claudin-15 promoter was identified. Abnormal gut barrier function may serve as a biomarker for the risk of IBD onset [111]. This study provides an explanation of reduced claudin-15 and VDR observed in the human IBD and deeper understanding of mechanisms by which tissue-specific VDR regulating barrier functions in the intestine.

### 3.2. Disrupted Intestinal Barrier in Colorectal Cancer (CRC)

The disruption of TJs is a common manifestation of CRC. Multiple factors, including host genetic background, immunity, environment, and microbiome contribute to the development of colon cancer [81,112,113]. At baseline, obesity, history of a sessile-serrated adenoma, and a family history of CRC were associated with claudin-1, occludin, and mucin-12 (MUC12) [114]. Mandle et al. investigated the effects of supplemental calcium (1200 mg, daily) and/or vitamin D_3_ (1000 IU daily) on intestinal barrier function-related biomarkers in 105 participants from a large colorectal adenoma recurrence chemoprevention clinical trial. They tested expression of claudin-1, occludin, and MUC12 in the normal-appearing colorectal mucosa. Following 1 year of treatment, in the calcium relative to the no-calcium group, the CLDN1, OCLD, and MUC12 expression increased. The estimated calcium treatment effects were greater among participants with baseline serum 25-OH-vitamin D concentrations below the median value of 22.69 ng/mL (claudin-1: 29%, *p* = 0.04; OCLD: 36%, *p* = 0.06; MUC12: 35%, *p* = 0.05), suggesting the link between vitamin D deficiency and TJ proteins in the human CRC. VDR is known to protect against tumorigenesis in the colon [4,115]. However, this study did not test the level of VDR in these patients. There are no data of microbiome from these patients.

We retrieved the expression of 20 claudins in normal and colon cancer patients by using Sequence Read Archive (SRA) microarray data (Gene Expression Omnibus (GEO) database GSE8671). Among these claudins, the expression of claudin-1, -2, -12, and -19 were increased in colon cancer patients; mRNA expression of claudin-5, -8, -15, -17, -20, and -23 were significantly decreased in patients with colon cancer; and the expression of claudin-3, -4, -6, -7, -9, -10, -11, -14, -16, and -18 did not show distinct changes in patients with colon cancer and normal. These data indicate the significance to study claudins in well-controlled experimental models and understand their different roles in the development of cancer.

Claudin-5 forms paracellular barriers and pores that determine permeability in the epithelia and endothelia. It is downregulated in colon cancer [116,117]. We found that reduction of colonic VDR expression is positively correlated with the low level of claudin-5 in patients with CRC [4]. We determined that VDR acts as a transcriptional regulator for the maintenance of physiological levels of the target gene *claudin-5* in the colon. Furthermore, in a conditional intestinal epithelial VDR-overexpressed mouse model, we found the protective role of VDR in the maintenance of TJs in the context of inflammation and colitis-associated colon cancer. High VDR level contributed to high claudin-5, reduced intestinal inflammation, and fewer tumors in the mouse colon, similar as the observation in the human CRC [4]. Interestingly, claudin-7 was not altered in the VDR-deficient colon, compared with the VDR+/+ colon in mice [4]. Our findings offer an insight into the VDR regulation on certain claudins, depending on the disease types.

VDR plays multiple functional roles of in the development of colon cancer [115,118]; thus, it is important to dissect the mechanisms by which VDR contributes to barrier function in protecting the host from tumorigenesis. The positively correlated status of VDR and claudins, such as claudin-5 and -15, could be potentially applied to risk assessment, early detection, and prevention of colitis and colitis-associated colon cancer. Hence, new insights into the mechanisms responsible for VDR and barrier dysfunction are needed. These studies will provide a new avenue to restore barrier functions and develop a novel protocol for risk assessment and prevention of CRC and other human cancers.

### 3.3. TJs and VDR in Lungs

Vitamin D/VDR signaling plays an important role in regulating the components of junctions and maintaining the integrity of epithelial barriers in multiple organs. In lungs, the permeability of the alveolar epithelial barrier is mainly regulated by the intercellular junctions [119,120]. In chronic inflammatory lung diseases, epithelial barrier function is impaired, disruption of the epithelial barrier resulted in alveolar permeability, i.e., paracellular movements of fluid from the interstitium to the pulmonary airspace, and infiltration of inflammatory cells [64]. Figure 2 shows well-organized ZO-1 in epithelial cells of the healthy lungs and lipopolysaccharide (LPS) treatment that disrupted the ZO-1 distribution in a mouse mode in vivo [65].

Vitamin D can promote epithelial barrier integrity or protect against epithelial barrier destruction in lungs. In the bronchial epithelial cell line 16HBE, vitamin D is able to counteract the effects of cigarette smoke extract–induced bronchial epithelial barrier disruption, such as transepithelial electrical resistance (TER) reduction, permeability increase, distribution anomalies, and increased cleavage of E-cadherin and β-catenin [63]. Vitamin D-treated BALB/c mice increased the expression of ZO-1 and E-cadherin, and reduced signs of asthma induced by toluene di-isocyanate [121]. Vitamin D treatment alleviated LPS-induced lung injury and preserved alveolar barrier function by inducing the expression of occludin and ZO-1 to maintain the pulmonary barrier in lungs [64,122], whereas in healthy mice fed with a vitamin D-poor diet indicated that vitamin D supplementation had little effect on epithelial integrity [123]. Vitamin D also promotes epithelial barrier function through its ability to increase expression of cystic fibrosis transmembrane conductance regulator in airway epithelial cells [124]. These studies indicate that vitamin D promotes the integrity and function of the epithelial barrier and protects against epithelial damage by dampening inflammatory response.

Lung VDR plays an important role in maintaining the pulmonary barrier integrity. We reported that VDR deletion increased lung permeability by altering the expression of TJ molecules, particularly claudin-2, -4, -10, -12, and -18 [65]. VDR KO mice showed significantly decreased expression of junctional proteins, including ZO-1, occludin, claudin-1, -2, -4, -12, -18, β-catenin, and VE-cadherin in lungs, compared with WT mice. claudin-2, -4, -12, and -18 mRNA and protein levels were significantly decreased in the lungs of VDR^−/−^ mice. However, VDR deletion did not change the mRNA levels of ZO-1, occludin, claudin-1, -3, -7, a-catenin, β-catenin, and VE-cadherin, suggesting the different mechanisms by which VDR regulates junctional proteins. These changes of TJs are closely correlated with airway epithelial barrier destruction in chronic pneumonia because of vitamin D/VDR deficiency [65]. Interestingly, *claudin-15* mRNA level was significantly decreased in lungs of VDR^−/−^ mice, but total claudin-15 protein was unchanged. Immunoblotting of claudin-15 protein in total lung tissue may not reflect its changes in certain types of lung cells [65]. Vitamin D supplementation alleviated LPS-induced lung injury and preserved alveolar barrier function through maintenance of the pulmonary barrier by inducing expression of occludin and ZO-1 in whole lung homogenates [64].

Several studies revealed the link of vitamin D deficiency to chronic inflammatory lung diseases, e.g., chronic obstructive pulmonary disease (COPD) and asthma [125,126,127,128]. COPD is a complex and progressive lung disease that is characterized by persistent airflow limitation resulting from chronic inflammation and structural changes [129]. The lower vitamin D levels and the more severe airflow obstruction were observed in patients with COPD [130] and accelerated decline in lung function [131]. VDR-deficient or knockout mice develop a COPD phenotype, whereas VDR overexpression can ameliorate inflammation in the lungs [132,133]. Vitamin D treatment reduces exacerbation rates in vitamin D-deficient patients with COPD or asthma, thus decreasing the incidence of acute respiratory tract infections [134,135,136].

### 3.4. Infection

The major obstacle *pathogens* must overcome is the intestinal epithelial barrier [137]. TJs are critical to providing host defense against pathogens. One of the beneficial roles vitamin D/VDR is to act as an important mediator of intestinal epithelial defenses against infectious agents. Vitamin D deficiency predisposes to more severe intestinal injury in an infectious model of colitis [138]. Vitamin D-deficient mice challenged with *C. rodentium* demonstrated increased colonic hyperplasia and epithelial barrier dysfunction. Vitamin D deficiency resulted in an altered composition of the fecal microbiome with or without *C. rodentium* infection [138], and 1,25(OH)_2_D_3_ altered *E. coli* O157:H7-induced reductions in TER, decreased permeability, and preserved barrier integrity in mice [139].

*Salmonella* are the causative agents of a variety of diseases ranging from diarrhea-generating gastroenteritis to systemic typhoid fever. *Salmonella* induces the disruption of TJs during infection. This is exemplified through decreases in TER, increases in tracer permeability, and TJ protein alterations in the infected intestinal cell lines and mouse tissues [137,140,141]. We and others have demonstrated that *Salmonella* targets TJ proteins [137,140,141], e.g., ZO-1 (Figure 3) [142,143,144], occludin [142,143,144], and claudins [28,105], and facilitates pathogenic enteric bacterial invasion [28]. The expression of claudin-7 is very stable in the colon in mouse colitis or bacterial infection [2,145].

Pathological bacterial translocation from the disrupted intestinal barrier leads to substantial complications and mortality in other organs, such as liver cirrhosis. Lee et al. [53] investigated the effects of calcitriol on bacterial translocation in cirrhotic rats. Cirrhotic rats were administrated with a 2-week course of active vitamin D_3_ (calcitriol, 0.1 µg/kg per day) or vehicle by oral gavage after thioacetamide (TAA) injection for 16 weeks. Vitamin D_3_ treatment significantly attenuated bacterial translocation and reduced intestinal permeability in TAA-induced cirrhotic rats. It upregulates the expressions of occludin in the small intestine and claudin-1 in the colon of cirrhotic rats directly independent of intrahepatic status. Vitamin D_3_ treatment also enriched *Muribaculaceae*, *Bacteroidales*, *Allobaculum*, *Anaerovorax*, and *Ruminococcaceae* in mice. This study showed that vitamin D_3_ attenuates intestinal leakage, reduces bacterial translocation, and enriches potentially beneficial gut microbiota, suggesting a potential therapeutic agent to prevent cirrhotic complications [53].

Vitamin D upregulates TJ proteins occludin and claudin-14 during *E. coli* urinary tract infection [73], likely improving the epithelial integrity, which in turn may ameliorate the protection against infection. This finding is relevant especially among patients with recurrent urinary tract infection (UTI) and where low vitamin D levels are anticipated. Bladder biopsies were obtained from post-menopausal women before and after a 3-month period of supplementation with 25-hydroxyvitamin D_3_ and ex vivo infected with *E. coli.* In biopsies, obtained before *E. coli* infection, vitamin D had no effect on TJ proteins. However, during *E. coli* infection, vitamin D induced occludin and claudin-14 in mature superficial umbrella cells of the urinary bladder. Vitamin D increased cell–cell adhesion, thus consolidating the epithelial integrity during infection [73].

We have started to collect more evidence of the tissue-specific role of VDR in intestinal health and inflammation. The studies in UTI [73] support the gender-difference and time-dependent (e.g., post-menopausal women) roles of VDR on TJs. We also noted gender differences in microbial metabolites [146] and virome in the mouse models [147]. We still do not have sufficient evidence for the gender- and time- specificity in the VDR-regulating barrier function in human samples.

The prevention of gut-barrier dysfunction is a viable approach for anti-infection, including COVID-19 [148]. Vitamin D deficiency is associated with adverse outcomes in infections [149]. There are compelling epidemiological associations between incidence and severity of COVID-19 and vitamin D deficiency [150]; one of the mechanisms is that the vitamin D pathway accelerates shutdown of T_H_1 cells in severe COVID-19. However, the mechanisms through VDR and tight junctions remain unknown. Even with the clear evidence that TJ function is influenced by pathogens, much more work is needed to understand the microbial regulation of intestinal TJs and to develop novel therapeutic targets for alleviating infection and infection-associated inflammation.

## 4. Elucidate Cellular and Molecular Mechanisms of Intestinal VDR in Regulating TJ Proteins

### 4.1. VDR Transcriptional Regulation of the Genes of TJ Proteins

Based on the genetic studies, VDR is a transcriptional factor regulating the genes of some TJ proteins, including claudin-2, -5, -12, and -15 in the intestinal epithelial cells [1,2,3,4]. For example, the sequence of functional vitamin D response element (AGATAACAAAGGTCA) is identified in the promoter of *claudin-5* gene. VDR deletion reduced the mRNA and protein levels of claudin-5 and upregulation of VDR increased the expression of claudin-5. These data demonstrate the genetic regulation of VDR on certain genes for TJs.

Another important question concerning VDR regulation is whether it is vitamin D dependent or independent. Although in normal cells with sufficient VDR expression, we observe the vitamin D-induced TJ expression through the VDR, the inflamed cells may use a totally different mechanism to alter TJs because the TJ proteins are also regulated by cytokines and other signaling pathways. In inflamed intestine of ulcerative colitis patients, VDR expression was low and claudin-2 was enhanced. Mechanistically, the enhanced claudin-2 promoter activity through the binding sites of nuclear factor kappa B (NF-κB) and signal transducer and activator of transcription (STAT) in inflamed cells without VDR [27]. Cytokines play an important role in the modulation of the intestinal epithelial TJ barrier. The pro-inflammatory cytokine, such as interleukin-6 (IL-6), interferon gamma (IFN-γ) and tumor necrosis factor alpha (TNF-α), induced increase in intestinal TJ permeability, which is an important pathogenic mechanism contributing to the development of intestinal inflammation [151]. IL-6 upregulates claudin-2 expression in intestinal epithelium [152]. On one hand, VDR is known to physically interact with NF-κBp65 and transcriptionally regulates IκBa to execute the anti-inflammatory role and suppress the inflammatory cytokines [153,154]. On the other hand, there is limited study on whether and how high inflammatory cytokines might directly suppress the expression of VDR.

### 4.2. Cellular Changes of TJs by 1,25(OH)_2_D_3_/VDR Status

TJ proteins, such as ZO-1, are upregulated in enterocytes by 1,25(OH)_2_D_3_ in vitro [1,43]. Severely disrupted TJs and increased permeability were seen in DSS-treated VDR^−/−^ colonic epithelial cells [43]. However, in the VDR^−/−^ mice, VDR deletion did not change the mRNA levels of ZO1, occludin, claudin-1, -3, -7, α-catenin, β-catenin, and VE-cadherin, suggesting the different mechanisms by which VDR regulates junctional proteins. We found that claudin-15 mRNA levels were significantly decreased in lungs of VDR^−/−^ mice, but total claudin-15 protein levels were not changed in the total lung tissue by Western blot. However, in the intestinal tissue, VDR deletion led to reduction of claudin-15 [3]. These results suggested that claudins have a different biological function in the different cells or tissues. Hence, further insights into the mechanisms responsible for intestinal VDR and barrier dysfunction are needed, especially in in vivo systems and human samples focusing on the tissue specificity and gender difference of VDR.

### 4.3. Other Regulators, e.g., Probiotics and Microtome, on TJs and VDR

Other factors, such as environmental factor and microbiome, also contribute to the changes of VDR and TJs. Most commensal organisms are kept segregated from the epithelium by the mucosa. Goblet cells are responsible for mucus production. Mucus consists of secretory immunoglobulin type A (sIgA) and AMPs (e.g., α-defensins), which are secreted by Paneth cells located in the small intestinal crypts and released to the colon. IgA regulates the composition and metabolic function of gut microbiota by promoting symbiosis between bacteria [155]. Macrophages in the underlying intestinal tissue produce milk fat globule-EGF factor 8, which directly targets IECs and regulates the integrity of intestinal barrier function [156]. The presence of these components helps to maintain the balance in the composition of the gut microbiota. Dysfunction in the production/secretion of individual proteins can lead to not only disturbances in the proportions of individual bacteria, but also the functions of barriers.

Whereas vitamin D has been intensively studied, the role of bacteria in modulating the effects of vitamin D and VDR signaling is not well known. We integrate our findings with other studies and, more importantly, understand how probiotics coordinate the effects of vitamin D/VDR and maintain the barrier function [157,158,159,160,161,162]. The novel information on the combined role of vitamin D and probiotics in anti-inflammation will contribute new concepts to the therapeutic methods for IBD and other inflammatory diseases. Although the potential importance of VDR as a therapeutic target has been recognized [163], no approach to date has safely and effectively altered its activity. Overuse of vitamin D to activate VDR signaling has potent hypercalcemic effect. In contrast, probiotic usage has little downside. Although probiotics can be dangerous if administered to very sick patients, the worst aspect is that probiotics will not produce any obvious benefit. By taking probiotics and/or vitamin D, it may indicate an inexpensive method to prevent chronic diseases.

## 5. Limits and Future Directions

We have made significant progress in the past decade to understand the role that vitamin D/VDR has in regulating tissue barriers. A series of molecular and biochemical experiments in vivo and in vitro, using VDR^−/−^ mice, conditional knockout models, organoids, and human database, allow us to investigate the in-depth mechanisms by which barriers are modulated in health and disease. However, the traditional model is to treat cultured cells or animals with vitamin D_3_. Our understanding of VDR in influencing intestinal barriers was limited to its role as a transcriptional factor and a receptor for vitamin D.

We now know the diverse roles and tissue specificity of VDR and its ligands [164,165]. We have begun to appreciate their clinical application in various human diseases [165]. For example, VDR ligands (e.g., calcipotriol) suppress pancreatitis and VDR acts as a master transcriptional regulator in pancreatic stellate cells [166]. VDR expressed in stroma from human pancreatic tumors and calcipotriol could reduce markers of inflammation and fibrosis in pancreatitis and human tumor stroma, suggesting vitamin D priming as an adjunct in the treatment of pancreatic ductal adenocarcinoma. In the kidney, 1,25(OH)_2_ VitD influences claudin-16-mediated Mg^2+^ transport [150]. However, in lungs, claudin-16 is very low and may not be necessary for respiratory function, or, other claudins may complement the function of claudin-16 in the lungs [167]. Thus, we need more studies on VDR’s tissue-, gender-, and time-specificity in regulating barrier function in health and disease. We need to consider sufficiently the physiological relevance in vivo and in disease models. We need to consider the other factors, e.g., microbiome and inflammatory triggers, which could take over in the vitamin D deficient context to alter TJ proteins, thus manipulating the barrier functions. We need to recognize that PCR data on mRNA alteration could not reflect the cellular distribution and functions of TJs. Functional studies and the syngenetic roles of TJ proteins are needed.

A series of molecular and biochemical experiments were performed in vivo and in vitro by using VDR transgenic mice and cultured human intestinal epithelial cells. However, the traditional model is to treat cultured cells or animals with vitamin D_3_. Our understanding of VDR in influencing intestinal barriers is limited to its role as a transcriptional factor and a receptor for vitamin D. In many clinical studies, the status of VDR was not considered in the study design or data interpretation. To move forward, special attention is needed on the roles of VDR in a gender-different and time-dependent manner, e.g., post-menopausal women, aging, and different stages of diseases.

Many researchers already notice the limits of cell lines. The recent development of organoids will help us to study tight junctions, physiological relevance, host–microbial interactions, and drug discovery [145,161,168,169,170,171,172]. As shown in Figure 4, claudin-7 is well maintained and distributes in the cell membranes in the mouse organoids. We found that TER of mouse colonoid-derived monolayers remained unchanged between the O-VDR overexerting colonoids and VDR^loxP^ colonoids [3]. We still need more studies on its tissue, gender, and time specificity in regulating barrier function. We need to consider sufficiently the physiological relevance of in vivo and in disease models, and the advancement of technologies and methods for the overall investigation of the intestinal barrier [173].

## 6. Conclusions

The recent progress reveals a novel activity of VDR in regulation of many tight junction proteins in primate cell structure and intestinal homeostasis and diseases (as shown in the Graphic Abstract). We aim to show the current state of knowledge on this topic and its potential therapeutic applications. This knowledge can be used to develop intestinal VDR-associated TJ proteins, e.g., claudin-5 and -15, as clinical biomarkers for identifying patients who may benefit from currently available interventions and could be used for the eventual development of novel strategies for the prevention and treatment of diseases. VDR signaling is also highly significant in regulating other proliferation and anti-inflammatory pathways [74,157,162,174]. We hope to integrate our findings with other studies and, more importantly, understand how the microbiome, probiotics, and metabolites coordinate the effects of vitamin D/VDR [146]. Our long-term goal is to develop individualized therapeutic strategies based on tight junction proteins [175] and vitamin D/VDR statuses for efficient and effective prevention and treatment of chronic diseases.

## Figures and Tables

**Figure 1 cells-11-01129-f001:**
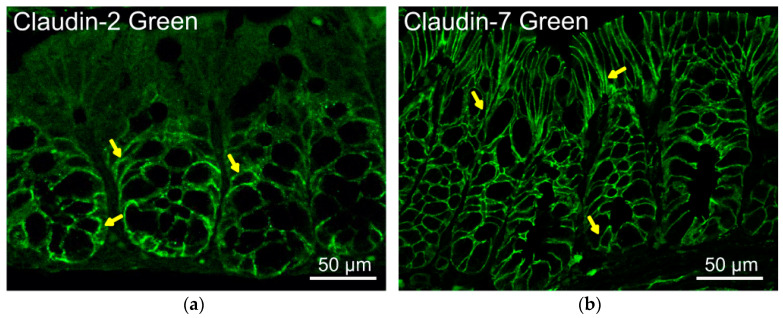
Location of claudin-2 and claudin-7 in normal mouse colon. (**a**) Normal claudin-2 distribution in cell membranes at the crypt base in the mouse colon (C57BL6/J mice, 6–8 weeks) (left panel: indicated by yellow arrows). (**b**) Normal claudin-7 distribution in cell membrane along crypt–villus axis in the mouse colon (right panel: indicated by yellow arrows) [28].

**Figure 2 cells-11-01129-f002:**
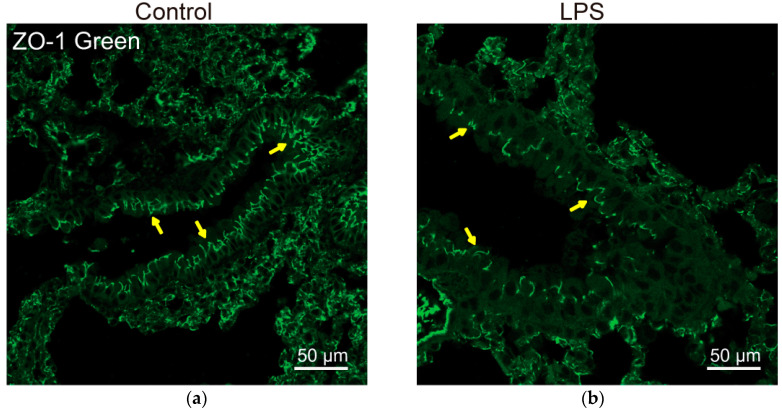
Disruption of tight junctions in mouse lung with LPS treatment. (**a**) Normal ZO-1 distribution in healthy mouse lungs (**left panel**: yellow arrow). (**b**) Reduced expression and disorganized structure of ZO-1 (**right panel**: yellow arrow) in the lungs of mice (C57BL6/J, 6–8 weeks) treated with LPS (10 mg/kg, intranasally). Mice were sacrificed after 24 h-infection [65].

**Figure 3 cells-11-01129-f003:**
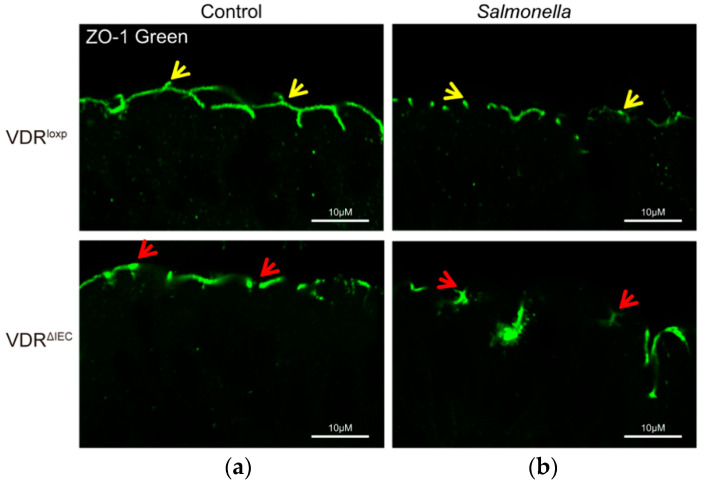
ZO-1 distribution before and after *Salmonella* infection in mouse colon. (**a**) ZO-1 distribution in the colon of VDR^loxp^ colon and VDR^ΔIEC^ mice. The yellow arrows indicate the “zipper” structure of ZO-1 in the apical side of the colon, and red arrows show the disrupted ZO-1 due to the intestinal epithelial VDR deletion in VDR^ΔIEC^ mice. (**b**) Disrupted ZO-1 distribution after *Salmonella* infection in the colon of VDR^loxp^ and VDR^ΔIEC^ mice. VDR^loxp^ and VDR^ΔIEC^ mice aged 6–8 weeks were infected with *Salmonella* and sacrificed 8 h post-infection, as described in our previous study [27,28].

**Figure 4 cells-11-01129-f004:**
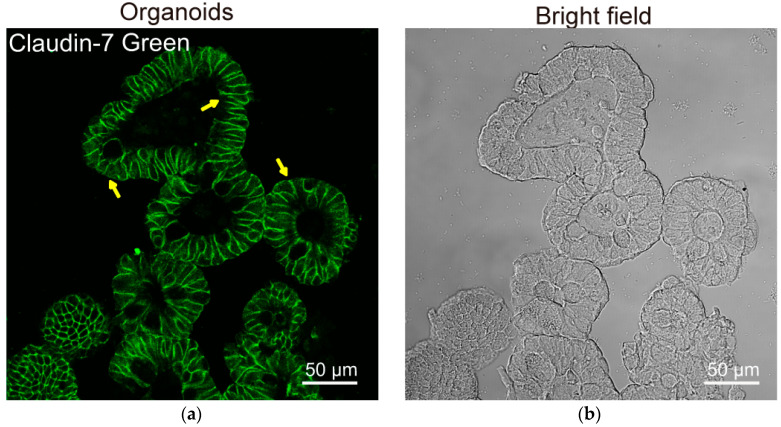
Distribution of claudin-7 in organoids. (**a**) Claudin-7 distribution in the cell membranes in the mouse organoids in fluorescent setting (left panel, yellow arrows). (**b**) Mouse organoids in the bright field. Experimental conditions are described in our previous report [145].

**Table 1 cells-11-01129-t001:** Vitamin D and VDR influence the expression and function of cell junction proteins.

Tissues & Cell Types	TJ Proteins	Cell Cultures In Vitro	Experimental Models In Vivo	Conclusions	References
Intestine	ZO-1 E-cadherin β-catenin	T84		The expression of VDR was positively correlated with the expression of ZO-1, occluding and claudin-5 in T84 cells.	[50]
		DLD-1		VDR acts as a regulator in the expression of intestinal mucosal barrier proteins under hypoxia environment. The expressions of VDR, ZO-1, occludin, claudin-1, and E-cadherin were obviously higher in vitamin D plus hypoxia group than in single vitamin D treatment group.	[51]
		SW480		1,25(OH)_2_D_3_ induced the expression of adhesion proteins and promoted the translocation of nuclear beta-catenin and ZO-1 to the plasma membrane.	[52]
	Occludin		Cirrhotic rats	Vitamin D_3_ treatment significantly attenuated bacterial translocation and reduced intestinal permeability in thioacetamide-induced cirrhotic rats. It upregulated the expressions of occludin in the small intestine and claudin-1 in the colon of cirrhotic rats directly independent of intrahepatic status. Vitamin D_3_ treatment also enriched Muribaculaceae, Bacteroidales, Allobaculum, Anaerovorax, and Ruminococcaceae.	[53]
		SW480-ADH		ROCK and MSK inhibition abrogates the induction of 1,25(OH)_2_D_3_ 24-hydroxylase (CYP24), E-cadherin, and vinculin, and the repression of cyclin D1 by 1,25(OH)_2_D_3_.	[54]
		Caco-2	VDR^−/−^ Mice	1,25(OH)_2_D_3_ enhanced TJs by increasing junction protein expression and TER and preserved the structural integrity of TJs in the presence of DSS. VDR knockdown reduced the junction proteins and TER. 1,25(OH)_2_D_3_ can stimulate epithelial cell migration in vitro.	[43]
	Claudins	Caco-2 SKCO15	VDR^−/−^ VDR^ΔIEC^ mice	1,25(OH)_2_D_3_ treatment upregulates claudin-2 expression in human epithelium cells. VDR deletion in intestinal epithelial cells led to significant decreased claudin-2 expression. CLDN2 gene is a direct target of the transcription factor VDR. VDR enhances claudin-2 promoter activity in a Cdx1 binding, site-dependent manner.	[2]
		Enteroids Caco-2 SKCO15	VDR^−/−^ Mice Salmonella- or DSS colitis model	In inflamed intestines of Salmonella- or DSS-induced colitis model, a lack of VDR regulation led to a robust increase of claudin-2 at the mRNA and protein levels post-infection. In inflamed intestines of ulcerative colitis patients, VDR expression was low and claudin-2 was enhanced. In inflamed VDR^−/−^ cells, 1,25(OH)_2_D_3_ enhanced claudin-2 promoter activity through the binding sites of NF-κB and STAT.	[27]
		SKCO15	O-VDR VDR^∆IEC^-Mice	Reduced claudin-15 was significantly correlated with decreased VDR in human IBD. O-VDR mice showed decreased susceptibility to chemically and bacterially induced colitis and marked increased claudin-15 expression in the colon. Correspondingly, colonic claudin-15 was reduced in VDR∆IEC mice, which were susceptible to colitis. Overexpression of intestinal epithelial VDR and vitamin D treatment resulted in significantly increased claudin-15. ChIP assays identified claudin-15 gene as a direct target of VDR.	[3]
		Human Colonoids SKCO15	VDR^−/−^ VDR^ΔIEC^-mice	Colonic VDR expression was low and significantly correlated with a reduction in claudin-5 in human CRC patients. Lack of VDR and a reduction of claudin-5 are associated with an increased number of tumors in the VDR^−/−^ and VDRΔIEC mice. CHIP assay identified CLDN-5 as a downstream target of the VDR signaling pathway.	[4]
			Human tissue microarr-ays	Vitamin D receptor (VDR) enhanced claudin-2 expression in colon and bile salt receptors VDR and Takeda G-protein coupled receptor 5 (TGR5) were highly expressed in esophageal adenocarcinoma and pre-cancerous lesions.	[23]
				1,25(OH)_2_D_3_ induces RANKL, SPP1 (osteopontin), and BGP (osteocalcin) to govern bone mineral remodeling; TRPV6, CaBP(9k), and claudin-2 to promote intestinal calcium absorption; and TRPV5, klotho, and Npt2c to regulate renal calcium and phosphate reabsorption.	[55]
		SW480-ADH		1,25(OH)_2_D_3_ activates the JMJD3 gene promoter and increases JMJD3 RNA in human cancer cells. JMJD3 knockdown or expression of an inactive mutant JMJD3 fragment decreased the induction by 1,25(OH)_2_D_3_ of several target genes and of an epithelial adhesive phenotype. It downregulated E-cadherin, claudin-1, and claudin-7.	[56]
			calbindin-D9k^−/−^ mutant mice	1,25(OH)_2_D_3_ downregulates cadherin-17 and upregulates claudin-2 and claudin-12 in the intestine, suggesting that 1,25(OH)_2_D_3_ can route calcium through the paracellular path by regulating the epithelial cell junction proteins.	[57]
		Caco-2	VDR^−/−^ mice	Claudin-2 and/or claudin-12-based TJs form paracellular Ca(^2+^) channels in intestinal epithelia. This study highlights a vitamin D-dependent mechanism in calcium homeostasis.	[1]
	Barrier and mucosal immunity		Rat intestine	The most strongly affected gene in intestine was CYP24 with 97-fold increase at 6 h post-1,25(OH)_2_D_3_ treatment. Intestinal calcium absorption genes: TRPV5, TRPV6, calbindin D(9k), and Ca(^2+^) dependent ATPase all were upregulated in response to 1,25(OH)_2_D_3_, However, a 1,25-(OH)_2_D_3_ suppression of several intra- and intercellular matrix modeling proteins, such as sodium/potassium ATPase, claudin-3, aquaporin 8, cadherin 17, and RhoA, suggesting a vitamin D regulation of TJ permeability and paracellular calcium transport. Expression of several other genes related to the immune system and angiogenesis was changed in response to 1,25(OH)_2_D_3_.	[58]
		Caco-2	C57BL/6 mice	1,25(OH)_2_D_3_ pre-treatment ameliorated the ethanol-induced barrier dysfunction, TJ disruption, phosphorylation level of MLC, and generation of ROS, compared with ethanol-exposed monolayers. Mice fed with vitamin D-sufficient diet had a higher plasma level of 25(OH)D_3_ and were more resistant to ethanol-induced acute intestinal barrier injury compared with the vitamin D-deficient group.	[59]
			VDR^ΔIEC^ VDR^ΔCEC^ Mice TNBS- colitis model	Gut epithelial VDR deletion aggravates epithelial cell apoptosis, resulting in mucosal barrier permeability increases.	[60]
		Caco-2	DSS-colitis model	1,25(OH)_2_D_3_ plays a protective role in mucosal barrier homeostasis by maintaining the integrity of junction complexes and in healing capacity of the colon epithelium.	[61]
Lung	E-cadherin β-catenin	A459		Vitamin D pre-treatment reduced TGF-β and Wnt/β-catenin signaling by increasing p-VDR, protected from E-cadherin degradation and led to the regression of EMT (epithelial–mesenchymal transition)-mediated myofibroblast differentiation.	[62]
		16HBE		Vitamin D is able to counteract the cigarette smoke extract-induced bronchial epithelial barrier disruption by TER (transepithelial electrical resistance) reduction inhibition, permeability increase, ERK phosphorylation increase, calpain-1 expression increase, and distribution anomalies and the cleavage of E-cadherin and β-catenin.	[63]
	ZO-1 Occludin		VDR^−/−^ WTMice	Vitamin D supplementation alleviated LPS-induced lung injury and preserved alveolar barrier function through maintenance of the pulmonary barrier by inducing expression of occludin and ZO-1 in whole lung homogenates.	[64]
	Claudins		VDR^−/−^ WT Mice	VDR^−/−^ mice showed significantly decreased ZO-1, occludin, claudin-1, claudin-2, claudin-4, claudin-10, β-catenin, and VE-cadherin expression in the lungs tissue compared with WT mice.	[65]
Kidney	ZO-1		Rat	Rat treated with 1,25(OH)_2_D_3_ is able to abrogate podocytes injury, detected as desmin expression and loss of nephrin and ZO-1.	[66]
	Claudins	HEK 293 OK	ICR mice	In kidney, 1,25(OH)_2_ VitD transcriptionally inhibits claudin-16 expression by a mechanism sensitive to CaSR and Mg^2+^. This renal effect of 1,25(OH)_2_ vitamin D may serve as an adaptive response to the 1,25(OH)_2_ vitamin D-induced increase in intestinal Mg^2+^ absorption.	[67]
Cornea	ZO-1 Occludin		VDR^−/−^ WT Mice	VDR^−/−^ mice showed the decreased expression of ZO-1 and occludin, and changed ZO-distribution on corneas compared with WT mice.	[68]
		Corneal epithelium	Mouse, Rabbit, Human	Vitamin D enhances corneal epithelial barrier function. Cells showed increased TER, decreased IP, and increased occludin levels when cultured with 25(OH)D_3_ and 1,25(OH)_2_D_3_.	[69]
Skin	ZO-1 Claudins		Human Psoriatic and Normal Skin	Psoriatic skin reduced VDR, ZO-1, and claudin-1 expression compared to normal skin, and showed a significant correlation of downregulated VDR expression to claudin-1 and ZO-1.	[70]
Brain	ZO-1 Occludin Claudins	End.3		Following hypoxic injury, 1,25(OH)_2_D_3_ treatment prevented decreased barrier function, and expression of zonula occludin-1, claudin-5, and occludin. VDR mediated the protective effect of 1,25(OH)_2_D_3_ against ischemic injury-induced blood–brain barrier dysfunction in cerebral endothelial cells.	[71]
Oral	ZO-1 E-cadherin β-catenin	HOK-16B		Vitamin D reinforces E-cadherin junctions by downregulating NF-κB signaling. In addition, vitamin D averts TNF-α-induced downregulation of the development of E-cadherin junctions in HGKs by decreasing the production of MMP-9, which was upregulated by TNF-α.	[72]
Urinary bladder	Occludin Claudin-14		Mouse, Human	During *E. coli* infection, vitamin D induced occludin and claudin-14 in mature superficial umbrella cells of the urinary bladder. Vitamin D increased cell–cell adhesion, thus consolidating the epithelial integrity during infection.	[73]

## Data Availability

Not applicable.

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
