# Peer review of "Vitamin D Receptor Influences Intestinal Barriers in Health and Disease"

_cells, 2022, doi:10.3390/cells11071129_

Round 1

Reviewer 1 Report

In this review paper the authors present an update of the literature on Vitamin D receptor and its role in some diseases. The paper appears to be confusing and not well-focused, there are some repetitions and it is sometimes difficult to follow the logic with which the data are presented. It is not always immediately clear if results are obtained in cell lines, mice or humans and how these are related to each other, and also if some of the results are original or taken from the literature. A figure/scheme summarizing the different activities and especially the targets of VDR would greatly help to put into context more easily the data.

If the figures are original new data, the authors should provide relevant experimental conditions. 

Table 1 should be reformatted, it is a comprehensive list of data each coming from a single reference but it is quite difficult to immediately pick the effects of VDR. Maybe the authors should try to summarize all the work in a more readable way.

All abbreviations should be defined the first time they are used. Please check/correct many typos and grammar errors.

Author Response

In this review paper the authors present an update of the literature on Vitamin D receptor and its role in some diseases. The paper appears to be confusing and not well-focused, there are some repetitions and it is sometimes difficult to follow the logic with which the data are presented. It is not always immediately clear if results are obtained in cell lines, mice, or humans and how these are related to each other, and also if some of the results are original or taken from the literature. A figure/scheme summarizing the different activities and especially the targets of VDR would greatly help to put into context more easily the data.

If the figures are original new data, the authors should provide relevant experimental conditions. 

Reply: We thank the reviewer for the constructive suggestions. We simplified our presentation and clarified the origins of data/results in our discussion, The four Figures are original new data. Per reviewer’s suggestion, we added more related experimental conditions and citations from our previous papers. 

Table 1 should be reformatted, it is a comprehensive list of data each coming from a single reference but it is quite difficult to immediately pick the effects of VDR. Maybe the authors should try to summarize all the work in a more readable way.

Reply:  we reorganized Table1 per reviewer’s suggestion.

All abbreviations should be defined the first time they are used. Please check/correct many typos and grammar errors.

Reply:  We define the abbreviations and add a list of abbreviations. We further checked and corrected typos and grammar errors.

Reviewer 2 Report

The review article entitled "Vitamin D Receptor Influences Intestinal Barriers in Health and Disease, submitted by Jun Sun and  Yong-guo Zhang for publication is an excellent  effort to compile the necessary information about VDR and their roles in different disease states.

The authors compiled information about colon cancer, infection,inflammatory bowel disease, and chronic inflammatory lung diseases. Although the authors did their efforts very positively but the manuscript require some corrections before publication.

  1. Authors must add one image showing the detailed mechanics of the receptor.
  2. The authors must add some information about the roles of VDR in gastrinomas.
  3. The authors similarly must discuss VDRs involvement in pateints underwent pancreatectomy
  4. The References are older, authors must update the complete bibliography.
  5. After addition of the data authors must improve the abstract, introduction, discussion and conclusion sections of the manuscript.
  6. The authors must check manuscript thoroughly for English language.

Author Response

Reviewer 2

The authors compiled information about colon cancer, infection, inflammatory bowel disease, and chronic inflammatory lung diseases. Although the authors did their efforts very positively but the manuscript require some corrections before publication.

  1. Authors must add one image showing the detailed mechanics of the receptor.

Reply:  We thank the reviewer for the constructive suggestions. We define the abbreviations and add a list of abbreviations.

2. The authors must add some information about the roles of VDR in gastrinomas.

Reply:  When we put three key words, “Vitamin D receptor, gastrinomas, and tight junctions” in PubMed and Google Scholar, we could not identify any related publications.

3. The authors similarly must discuss VDRs involvement in pateints underwent pancreatectomy

Reply:  VDR is involved in patients underwent pancreatectomy. However, we did not find mechanistic research papers on tight junctions and VDR in pancreatectomy by searching PubMed and Google Scholar. The title of current review is “Vitamin D Receptor Influences Intestinal Barriers in Health and Disease”. Therefore, we focus on the intestinal diseases.  We apologize if we missed critical papers in the field. We would appreciate reviewers provide us with the DOI of the related paper.

4. The References are older, authors must update the complete bibliography.

Reply: The references are mixed with old and recent publications. We tried to update bibliography with the papers published in 2022.

5. After addition of the data authors must improve the abstract, introduction, discussion and conclusion sections of the manuscript.

Reply: We revised our abstract, introduction, discussion and conclusion sections accordingly in the revised manuscript.

6. The authors must check manuscript thoroughly for English language.

Reply: We further checked and corrected typos and grammar errors.

Reviewer 3 Report

This article reviewed the effects of vitamin D receptor on the alteration of epithelial tight junction in inflammatory bowel diseases, colon cancer, chronic obstructive pulmonary disease

The author described that there was the insufficient evidence on tissue-, gender, and time-specificity in the introduction. I agreed with authors’ comments and the reader will expect discussion for this part to be explained in the manuscript body. Tissue-specific responses was described in the intestine and lung; however, the explanation of the gender or time-specific response did not summarize in the manuscript. Please add commets on this manuscript body. If not. please describe this part only in Future Direction.

Minor point

  1. abstract : Please remove the subtitle according to the author instructions

Page 3. line 114, clarify “_he”

Page 4 line 4140, Please clarify the order of the references.

Page 4 line 153, Please clarify “Zo-1  1”

Page 5 line 190, “SRA” please, describes whole words

Page 8 line 132, Please, describe the association between the COVID-19 & tight junction molecule

Page 8 line 353 ~ page 9 356

Page 8, line 332, Please, describe the association between the COVID-19 & tight junction molecule

Page 8, line 353 ~ Page 9, line 356. The effect of cytokine on TJ molecule expression seems to be well known. I wonder how the cytokines affect on the VDR expression

Page 10 line 406, 408 please check the reference format

Page 11. line 422 mucrial -> microbial ?

Page 11. line 446 colnioids -> colonoid?

Author Response

Reviewer 3

The author described that there was the insufficient evidence on tissue-, gender, and time-specificity in the introduction. I agreed with authors’ comments and the reader will expect discussion for this part to be explained in the manuscript body. Tissue-specific responses was described in the intestine and lung; however, the explanation of the gender or time-specific response did not summarize in the manuscript. Please add comments on this manuscript body. If not. please describe this part only in Future Direction.

Reply: We thank the reviewer for the constructive suggestions. We added the explanation of the gender or time-specific response in the revised manuscript. We also describe this part only in the revised Future Direction.

Minor point

  1. abstract: Please remove the subtitle according to the author instructions

Reply: We removed the subtitles

Page 3. line 114, clarify “_he”. Yes, we corrected it.

Page 4 line 4140, Please clarify the order of the references.

Yes, we corrected it.

Page 4 line 153, Please clarify “Zo-1  1”: We removed the extra “1”.

Page 5 line 190, “SRA” please, describes whole words:

Yes, we added the whole word.

Page 8 line 132, Please, describe the association between the COVID-19 & tight junction molecule

Page 8 line 353 ~ page 9 356

Page 8, line 332, Please, describe the association between the COVID-19 & tight junction molecule

Yes, we added discussion on COVID-19 & barriers and role of Vitamin D in COVID (page. 9).

Page 8, line 353 ~ Page 9, line 356. The effect of cytokine on TJ molecule expression seems to be well known. I wonder how the cytokines affect on the VDR expression

The inflammation caused by inflammatory cytokines could reduce the VDR expression. However, we do not know the direct effect of cytokines on the VDR expression. We added related discussion in the revised manuscript (page 9).

Page 10 line 406, 408 please check the reference format

Yes. We checked.

Page 11. line 422 mucrial -> microbial ? Yes.

Page 11. line 446 colnioids -> colonoid?  Yes. We corrected all the indicated typos.

Round 2

Reviewer 1 Report

In this revised version of the paper, the authors have addressed my previous comments. I still think that Table 1 is quite difficult to read but the paper is improved.

Author Response

Rev. 1

In this revised version of the paper, the authors have addressed my previous comments. I still think that Table 1 is quite difficult to read but the paper is improved.

Reply: Thank you. We further modified Table 1 by adding the categories based on tissue resource, TJs proteins studied, experimental models (in vitro and in vivo), conclusions, and references. We hope to help readers to summarize the progress in the field and identify papers for further reading. It will be helpful if Reviewer 1 could provide detailed suggestions for us to improve this table.

Reviewer 2 Report

Although the Authors revised manuscript, but the first two queries have not been addressed  properly,

  1. Authors should add one image showing the detailed mechanics of the receptor.

"The authors replied instead about the abbreviations, authors must add one figure atleast to make their work elegant and enhance the readability".

  1. The authors should add some information about the roles of VDR in gastrinomas.

The authors did not try to search the articles,

Please refer,

  1. cell, 2014, doi: 10.1016/j.cell.2014.08.007 title " Vitamin D receptor-mediated stromal reprogramming suppresses pancreatitis and enhances pancreatic cancer therapy

  1. Daniel Bikle, John Adams, and Sylvia Christakos,Chapter 28. Vitamin D: Production, Metabolism, Mechanism of Action, and Clinical Requirements

Author Response

Rev. 2

  1. Authors should add one image showing the detailed mechanics of the receptor.

"The authors replied instead about the abbreviations, authors must add one figure at least to make their work elegant and enhance the readability".

Reply: Sure. In R1, we submitted a graphic abstract, which was not shown in the word file. In the current revision, we add the graphic abstract in the Text (Page 12).

  1. The authors should add some information about the roles of VDR in gastrinomas.

The authors did not try to search the articles,

Please refer,

  1. cell, 2014, doi: 10.1016/j.cell.2014.08.007 title " Vitamin D receptor-mediated stromal reprogramming suppresses pancreatitis and enhances pancreatic cancer therapy
  2. Daniel Bikle, John Adams, and Sylvia Christakos, Chapter 28. Vitamin D: Production, Metabolism, Mechanism of Action, and Clinical Requirements.

Reply: Thank you. We added these two papers as suggested and discussed the roles of VDR in gastrinomas (Page 10).

Reviewer 3 Report

The author responded appropriately.

Author Response

Thank you.